# Inverse Association of Fruit and Vegetable Consumption with Nonalcoholic Fatty Liver Disease in Chinese Patients with Type 2 Diabetes Mellitus

**DOI:** 10.3390/nu14214559

**Published:** 2022-10-29

**Authors:** Lin-Jia Du, Zhi-Ying He, Xiao Gu, Xiang Hu, Xing-Xing Zhang, Li-Juan Yang, Jing Li, Lin-Yu Pan, Ying-Qian Li, Bo Yang, Xue-Jiang Gu

**Affiliations:** 1Department of Endocrine and Metabolic Diseases, 1st Affiliated Hospital of Wenzhou Medical University, Wenzhou 325000, China; 2School of Public Health and Management, Wenzhou Medical University, Chashan University Town, Wenzhou 325015, China

**Keywords:** NAFLD, dietary, type 2 diabetes mellitus, fruit and vegetable

## Abstract

We aimed to investigate the association of fruit and vegetable consumption with nonalcoholic fatty liver disease (NAFLD) in Chinese patients with type 2 diabetes mellitus (T2DM). This cross-sectional study included 2667 Chinese patients with T2DM aged 18 to 76 years from March 2017 to October 2021. Dietary intake was assessed using a food frequency questionnaire, and prevalent NAFLD was diagnosed with abdominal ultrasonography. High fruit–vegetable consumption was determined using ≥500 g/day consumption of both fruit and vegetable, and both fruit and vegetable consumption were divided into three categories of <200 g/day (low), 200–400 g/day (median) and >400 g (high). The primary outcome measurement was multivariate-adjusted odds ratios (ORs) with 95% confidence intervals (CIs) for the prevalence of NAFLD in relation to the highest fruit and (or) vegetable intake compared with the lowest. Secondary analyses were conducted to assess the effects of either fruit or vegetable intake on the fatty liver index (FLI) using multivariable linear regressions. There were 1694 men and 973 women in this study, and 1445 (54.06%) participants had prevalent NAFLD. Patients with high fruit–vegetable intake had a lower prevalence of NAFLD than those with low fruit–vegetable intake (52.04% vs. 56.48%), but this difference was not statistically significant (*p* = 0.065). Vegetable intake had a significantly inverse association with NAFLD (OR: 0.68, 95% CI: 0.52–0.90), but this association was not pronounced with fruit intake (OR: 1.23, 95% CI: 0.89–1.69) or fruit–vegetable intake (OR: 0.90, 95% CI: 0.73–1.10). Additional analyses showed that an increase in vegetable intake was linearly associated with a significant reduction in FLI (β: −1.028, 95% CI: −1.836, −0.219). In conclusion, higher vegetable consumption was associated with lower odds of NAFLD in Chinese patients with T2DM, which suggested that increased vegetable intake might protect patients with diabetes against NAFLD.

## 1. Introduction

Diabetes is a metabolic disorder characterized by chronic hyperglycemia caused by impaired insulin secretion or impaired insulin action or both. Metabolic disorders such as type 2 diabetes mellitus (T2DM), obesity and fatty liver disease often coexist in patients with insulin resistance. As a common metabolic disease, nonalcoholic fatty liver disease (NAFLD) has become the most common chronic liver disease worldwide. The global prevalence of NAFLD is 25.24% [1]; in China, the prevalence of NAFLD is 29.2% among adults aged more than 18 years old [2]. Given that NAFLD is becoming a nationwide health burden that increases cardiometabolic risk, how to treat NAFLD is a public health problem that deserves priority attention. In addition to clinical pharmacological treatments, lifestyle interventions such as energy restriction, dietary changes, and physical activity increase also play essential roles in NAFLD therapies. Plant-based, high-fiber and low-fat diets have long been recommended to manage some chronic diseases, such as cardiovascular disease, T2DM and obesity [3]. To date, many studies investigated whether fruit and/or vegetable intake affect NAFLD, but their results were inconsistent. A cross-sectional study investigating 26,891 adults in China showed that high intake of green leafy vegetable was inversely associated with NAFLD, particularly in female and nonobese participants [4]. Similar findings were also seen in another study among 52,280 Korean adults, and their data indicated that fruit intake, vegetable intake and total fruit–vegetable intake were separately associated with reduced NAFLD risks [5]. Nevertheless, a nested case–control analysis of an ethnically diverse population cohort in America showed no significant association between vegetable or fruit intake and NAFLD [6]. Therefore, whether or how fruit or vegetable intake influences NAFLD risk is currently uncertain.

NAFLD is closely linked to obesity and interrelated with insulin resistance in T2DM. It is well known that NAFLD and diabetes often coexist, and the prevalence of NAFLD could be as high as 49~62% in patients with T2DM [7]. NAFLD and T2DM not only share common risk factors but also act synergistically to increase the risk of adverse clinical outcomes (intra-and extra-hepatic) when they occur together [8,9]. Laboratory data to date suggest that some potential mediators are involved in glucose and lipid metabolism in determining the interrelation between NAFLD and T2DM. The secondary bile acids interact with various nuclear receptors such as farnesoid X receptor (FXR) and Takeda G protein-coupled membrane receptor 5 (TGR5). The activation of TGR5 in enterocytes increases the release of glucagon- like peptide 1 (GLP1), which increases insulin secretion from pancreatic islet beta cells, thereby affecting plasma glucose concentrations [10]. The activation of hepatic FXR receptors decreases fatty acid and triglyceride synthesis due to potentially decreasing hepatic lipid accumulation [11].

Most of the previous epidemiological studies have been conducted to examine the effects of fruit and/or vegetable intake on NAFLD risk in the general population and have not targeted the population with diabetes. Given the biological interrelation between NAFLD and T2DM, the novelty of our study is our examination of the association of fruit and vegetable intake with prevalent NAFLD among the population with diabetes in China.

## 2. Materials and Methods

### 2.1. Study Population

This cross-sectional study was based on the baseline analyses nested within the Metabolic Management Center (MMC) cohort at the 1st Affiliated Hospital of Wenzhou Medical University. The MMC is a standardized management system for metabolic diseases managed by the National Clinical Research Center for Metabolic Diseases. Participants in the MMC cohort were recruited from inpatients and outpatients with chronic metabolic diseases, mostly with diabetes. Laboratory results and special examination results of the patients were collected. All participants were asked to complete a detailed questionnaire including items regarding personal information, dietary intake, lifestyle factors and health condition at baseline. As shown in the flow diagram, this study used baseline data from 3510 participants with T2DM who were enrolled in the MMC cohort from March 2017 to October 2021. After excluding participants who lacked abdominal ultrasound results (*n* = 167), were missing essential data (*n* = 13), had significant alcohol consumption (*n* = 468), and had viral hepatitis (*n* = 185) or other liver diseases (*n* = 10), the final study sample comprised 2667 participants. The study population flow chart is described in Figure 1.

### 2.2. Diagnosis

NAFLD was diagnosed by abdominal ultrasound. Those with significant alcohol consumption (≥210 g/week for men, ≥140 g/week for women) were defined as overdrinkers and excluded [12,13]. Other liver diseases (chronic hepatitis B or C, autoimmune liver diseases, cirrhosis or liver cancer, etc.) were identified by their clinical examination results (e.g., positive hepatitis B or C serology) or their self-reported or physician-reported medical history.

Hypertension was defined as a systolic blood pressure (SBP) ≥130 mmHg, a diastolic blood pressure (DBP) ≥85 mmHg or receiving antihypertensive treatment. Diabetes was defined as a fasting blood glucose (FBG) level ≥7.0 mmol/L, a 2 h postprandial blood glucose (PBG)/random blood glucose level ≥11.1 mmol/L or using antidiabetic medication. Hyperlipidemia was defined as a triglyceride (TG) level ≥2.3 mmol/L, total cholesterol (TC) level ≥6.2 mmol/L, low-density lipoprotein cholesterol (LDL-c) level ≥4.1 mmol/L or high-density lipoprotein (HDL) level <1.0 mmol/L [14]. Overweight/obesity was defined as a BMI ≥24 kg/m^2^ [15]. Central obesity was defined as a waist circumference (WC) ≥90 cm for men and ≥85 cm for women [14].

### 2.3. Diet and Covariate Assessments

Dietary data were collected using the seven-item food frequency questionnaire (FFQ). Participants were asked about their daily consumption of vegetables, fruits, bean products, and salt, as well as their weekly intake of fish and sugary beverages.

The sociodemographic variable and health condition data, including age, sex, educational attainment, current smoking status, current drinking status, physical activity level, course of diabetes and family history of diabetes, were collected from the questionnaires that the participants completed at baseline. Physical activity included the amount of activity at work and at rest; those who performed light physical labor (e.g., office work) and were inactive at rest were considered almost never active.

Anthropometric parameters, including height, weight, waist circumference (WC), hip circumference (HC), and blood pressure, were measured by a trained nurse. Body mass index (BMI) was calculated by dividing body weight (kg) by the square of height (m^2^).

Blood samples of the participants were analyzed for blood glucose, insulin, glycosylated hemoglobin (HbAc1), uric acid (UA), blood lipid (including triglycerides (TG), total cholesterol (TC), low-density lipoprotein cholesterol (LDL-c) and high-density lipoprotein cholesterol (HDL-c), alanine aminotransferase (ALT), aspartate aminotransferase (AST) and γ-glutamyl transpeptidase (GGT) levels with an automatic biochemical analyzer (AU 5800, Beckmann Coulter, Brea, CA, USA) and an HbA1c analyzer (Premier Hb9210, Trinity Biotech, Bray, Wicklow, Ireland). HOMA-IR was calculated as follows: HOMA-IR = [fasting plasma insulin concentration (mU/L) * fasting plasma glucose (mmol/L)]/22.5. The fatty liver index (FLI) was calculated as follows: FLI = [e^(0.953 × ln(TG) + 0.139 × BMI + 0.718 × ln(GGT) + 0.053 * WC − 15.745)]/[1 + e^(0.953 * ln(TG) + 0.139 × BMI + 0.718 × ln(GGT) + 0.053 * WC − 15.745) * 100] [16].

### 2.4. Statistical Analyses

Participants’ characteristics are presented as the mean (standard deviation) for continuous variables with normal distribution and the median (interquartile range) for skewed-distribution variables, while categorical data are presented as numbers and percentages. According to the daily recommendation of fruit–vegetable intake, participants were classified into two groups of ≥500 g/day intake (high) vs. <500 g/day intake categories to compare the baseline characteristics between the two groups. The between-group differences were examined by T tests for continuous data with normal distribution and log-transformed normal distributions, Mann–Whitney U tests for ordinal categorical variables and chi-square tests for dichotomous variables. If covariate information was missing (<20%), we used multiple imputations based on five replications to account for missing data.

Either fruit or vegetable consumption was divided into three categories of <200 g/day (low), 200–400 g/day (median) and >400 g/day (high). Primary measurements were odds ratios (ORs) and 95% confidence intervals (CIs) for the fruit and/or vegetable intake in relation to prevalent NAFLD, calculated using the logistic regression models with multiple adjustments for demographic factors, lifestyle factors, dietary factors, and biochemical indices, with the lowest category of fruit and/or vegetable intake as the reference. We fitted four models with different covariates to adjust for potential confounders. Model 1 was a crude model. Model 2 was adjusted for demographic parameters (age, sex, educational attainment), anthropometric parameters (BMI, WC, HC, BP) and diabetes duration. Model 3 was additionally adjusted for family history, lifestyle factors (smoking, drinking, physical activity level) and food intake including bean products, salt, fish and sugary beverages). Model 4 was adjusted for the variables included in Model 3 plus biochemical indices (HbA1c, ALT, AST and serum lipid levels). In secondary analyses, multivariable linear regression models with adjustments for demographic parameters, diabetes duration, anthropometric parameters, diabetes family history, lifestyle factors, food intake and the biochemical indices were applied to assess the potential effects of fruit and vegetable intake on FLI. Statistical analyses were performed using SPSS for Windows, version 25 (IBM, Armonk, NY, USA), and *p* values < 0.05 were considered statistically significant.

### 2.5. Ethical Approval

The data used in this study are from a Metabolic Management Center (MMC) cohort at the 1st Affiliated Hospital of Wenzhou Medical University. The Review of Ethics Committee in Clinical Research (ECCR) of the First Affiliated Hospital of Wenzhou Medical University approved this study on 22 November 2021 with the project name Serological, metabolomic, and genomic studies of metabolic diseases and issuing number KY2021-173.

## 3. Results

### 3.1. Baseline Characteristics

Baseline characteristics of the 2667 participants between high and low fruit–vegetable intake groups are presented in Table 1. This study included 1694 men and 973 women with an average age of 49.94 years, and NAFLD was diagnosed in 1445 (54.06%) participants. The high fruit–vegetable intake group showed a lower prevalence of NAFLD than the low fruit–vegetable intake group (52.04% vs. 56.48%), but the difference was not significant (*p* = 0.065). Patients with high fruit–vegetable intake were likely to be older, leaner and more physically active, to smoke less and to have a longer course of diabetes.

### 3.2. Fruit and Vegetable Intake and NAFLD

As shown in Table 2, high fruit–vegetable intake was not significantly associated with low odds of NAFLD (multivariate-adjusted OR: 0.90, 95% CI: 0.73–1.10, *p* = 0.29). Vegetable intake showed a negative association with NAFLD after multiple adjustments (OR: 0.68, 95% CI: 0.52 to 0.90, *p* = 0.007), and a 200 g increase in vegetable intake reduced the odds of NAFLD by 15% (OR: 0.85, 95% CI: 0.75 to 0.96, *p* for trend = 0.008). However, there was no significant association between fruit intake and NAFLD prevalence (OR: 1.23, 95% CI: 0.89 to 1.69, *p* = 0.20).

### 3.3. Subgroup Analyses

Analyses stratified by sex, age, diabetes duration (median), overweight/obesity, central obesity, hypertension and hyperlipidemia were performed to test the potential interactions of the cardiometabolic risk factors with vegetable intake on the prevalent NAFLD (Table 3). Neither predefined subgroup showed a significant impact on NAFLD.

### 3.4. The Effects of Fruit and Vegetable Intake on FLI

The associations between fruit and vegetable intake and FLI are reported in Table 4. Vegetable intake showed a negative association with FLI (β: −1.028, 95% CI: −1.836 to −0.219, *p* = 0.013), but this association was not found to be statistically significant with fruit intake (β: −0.255, 95% CI: −1.137 to 0.627, *p* = 0.571).

## 4. Discussion

In this cross-sectional study, our primary analyses showed that vegetable intake, rather than fruit intake, was inversely associated with NAFLD, which was not affected by other cardiometabolic risk factors such as central obesity, hypertension or hyperlipidemia. In additional analyses, vegetable intake was linearly associated with a reduced degree of lipid accumulation, as assessed by FLI. These findings suggested that vegetable intake might have an independent role in preventing NAFLD.

In previous studies, certain dietary patterns rich in fruit and vegetables have been shown to play a protective role in metabolic diseases. The Mediterranean diet (MeD), which is characterized by the high consumption of extra virgin olive oil, nuts, red wine, vegetables and other polyphenol-rich elements, has been proven to be associated with a lower risk of metabolic diseases via greater insulin resistance [17]. Dietary Approaches to Stop Hypertension (DASH) is a dietary pattern initially designed for the control of hypertension, characterized by a diet rich in fruits, vegetables and low-fat dairy products and low in meats and sweets that has been confirmed to have a positive influence on metabolic syndrome control and weight reduction [18,19]. Many studies have verified that NAFLD, as a common metabolic disease, could benefit from dietary patterns rich in fruits and vegetables, such as the MeD, DASH and plant-based diets. They were proven to reduce hepatic lipid accumulation, liver stiffness and the likelihood of fatty liver in overseas studies [20,21,22]. A study conducted in Guangzhou, China, also showed a markedly lower prevalence of NAFLD associated with Chinese adults who adhered to DASH, especially in women and those without abdominal obesity [23]. The above studies suggested that fruits and vegetables were protectors of NAFLD in the general population. In concordance with previous studies, the protective effect of vegetables was also present in patients with diabetes in our study.

Several mechanisms might partly explain the inverse association between vegetable intake and NAFLD. Antioxidants found in vegetables, such as vitamin D, E and astaxanthin, can decrease the accumulation of reactive oxygen species and then reduce insulin resistance, inflammation and fibrogenesis, eventually ameliorating the disease [24,25]. Some micronutrients, such as magnesium, found in vegetables may also play a role. Mg improved insulin sensitivity and glucose tolerance in an animal model by stimulating the expression and translocation of the GLUT4 gene, as well as suppressing the gluconeogenesis pathway and glucagon receptor gene expression [26]. Another animal study also demonstrated that magnesium deficiency was involved in NAFLD by hepatic inflammatory processes and hepatocyte enlargement [27]. Epidemiological studies have indicated that dietary fiber, a nondigestible carbohydrate derived from plant-based food, was found to be beneficially associated with cardiometabolic disorder including NAFLD in general populations. [28,29]. Prebiotics, which are special kinds of dietary fiber, have been confirmed to reduce plasma lipid and hepatic TG concentrations by reducing de novo fatty acid synthesis, improving insulin resistance and regulating the gut microbial community [30]. Dietary fiber can also be fermented by gut microbes and generate short-chain fatty acids (SCFAs) such as acetate and butyrate [31] that can inhibit oxidative stress and inflammation by activating AMP-activated protein kinase (AMPK) [32,33]. Moreover, it can reduce the frequency of eating by intensifying satiety, thus restricting energy intake and promoting weight loss [34].

Interestingly, compared with vegetable intake, fruit intake did not show a significant association with NAFLD. The effect of fruit intake on NAFLD is still under debate. An inverse association was found between the intake of fruits and the likelihood of having NAFLD in Hong Kong Chinese individuals in a cross-sectional study [35]. A Japanese cross-sectional study showed no obesity-independent association between fruit intake and NAFLD in a middle-aged population [36]. Another study recruited 27,214 adults in China and demonstrated that orange intake was positively associated with the prevalence of NAFLD [37]. A case–control study confirmed that a high fruit diet was a potential risk factor for NAFLD in Lebanese patients [38]. Different kinds of fruit might have different influences on metabolic diseases. A cohort study of Asian adults showed that the higher consumption of tropical or higher-GI fruit was related to a higher risk of T2DM, while a higher intake of temperate fruit was associated with a lower T2DM risk in women [39]. We did not distinguish the specific kinds of fruit in our study, which might have affected results. On the other hand, the effect of fruit might be related to the amount of intake. The 2021 dietary guidelines for Chinese residents suggest an intake of 200 g~350 g of fruit per day to reduce the risk of some chronic diseases. In our study, conducted with patients with diabetes, most patients did not reach the standard (69.88% in the <200 g/d group).

In contrast with the previous studies in general populations, our study was conducted in a population with diabetes. Given that there are notable differences in eating habits between patients with diabetes and the general population, the conclusions drawn from the studies of general population may not be applicable to the population with diabetes. In addition, to date, there is no study specifically focusing on the effect of fruit and vegetable intake on NAFLD in diabetes in China. Some limitations of this study should be noted. First, liver biopsy is the gold standard for fatty liver diagnosis. Our research used abdominal ultrasound to diagnose fatty liver, which was dependent on the subjective judgement of the examining doctors and had low sensitivity for the diagnosis of mild fatty liver [40]. However, because of its invasiveness, biopsy is difficult to use for screening in the setting of large populations. Ultrasound by contrast was an acceptable screening option. Second, the dietary data were collected by self-report; therefore, recall bias was inevitable. Third, medication use (e.g., metformin, GLP-1) might modify the diet-NAFLD association, but we did not examine the effects of medicine. Fourth, we used a validated questionnaire combined with a seven-item food intake inventory, more comprehensive information on total calories or other individual foods were not obtained. Therefore, the possibility cannot be excluded that residual confounders from unknown factors such as a specific dietetic pattern may have caused a potential change in association directions. Finally, this was a cross-sectional study and was therefore not able to reflect the causal relationship between fruit and vegetable intake and NAFLD. The association should be validated by long-term follow-up studies. In future studies, we will recruit more participants to conduct a follow-up study, which may help validate the prospective association between fruit and vegetable intake and the incident NAFLD.

## 5. Conclusions

Vegetable intake was inversely associated with prevalent NAFLD in Chinese T2DM patients, while fruit intake had no significant association with NAFLD. For Chinese patients with T2DM, increased intake of vegetable may be protective against NAFLD. These findings may have provided new insights into an evidence-based reference for supporting vegetable intake in dietary structure among Chinese patients with diabetes.

## Figures and Tables

**Figure 1 nutrients-14-04559-f001:**
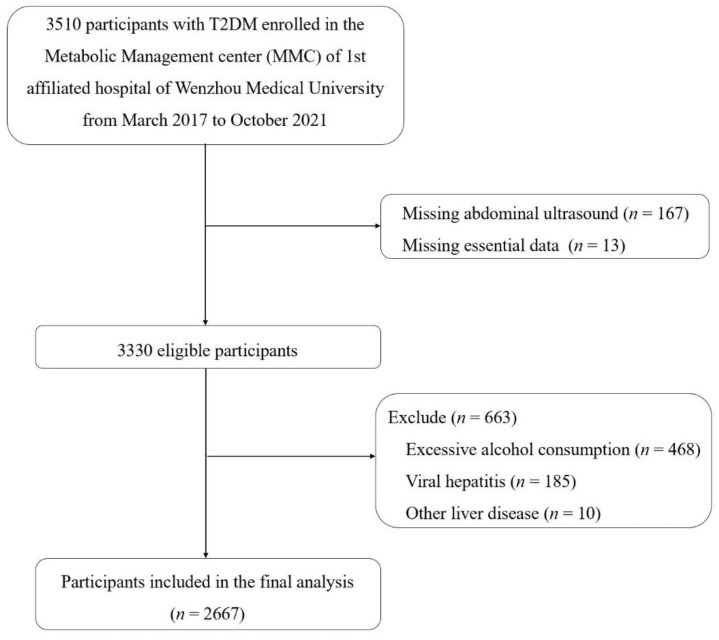
Flow diagram of study participant selection.

**Table 1 nutrients-14-04559-t001:** Baseline characteristics of the participants according to their fruit–vegetable intake.

	Total	<500 g/day	≥500 g/day	*p*
Case/participants (%)	1445/2667 (54.06)	714/1277 (56.04)	731/1369 (52.48)	0.065
Sex (male, %)	63.52	65.23	61.96	0.080
Age (years)	49.94 (11.79)	49.14 (12.21)	50.68 (11.34)	0.001 *
Diabetes duration (month)	64 (4–136)	60 (3–128)	69 (5–146)	0.004 *
Family history of diabetes (%)	56.62	56.35	56.88	0.783
**Anthropometric parameters**				
DBP (mmHg)	75.01 (11.02)	75.49 (11.30)	74.57 (10.73)	0.031 *
SBP (mmHg)	126.5 (18.59)	126.55 (18.81)	126.45 (18.40)	0.892
BMI (kg/m^2^)	24.55 (3.70)	24.70 (3.79)	24.40 (3.60)	0.039 *
WC (cm)	88.43 (10.15)	88.95 (10.30)	87.95 (9.98)	0.012 *
HC (cm)	94.42 (7.45)	94.69 (7.47)	94.17 (7.43)	0.073
**Lifestyle factors**				
Educational attainment (≥12 year%)	26.30	26.81	25.84	0.571
Current smoking (%)	29.70	33.20	26.50	<0.001 *
Current drinking (%)	34.19	34.69	33.74	0.604
Physical activity (almost never, %)	27.42	29.49	25.52	0.022 *
**Fruit and vegetable intake (%)**				
Fruit intake				
<200 g/day	69.88	79.56	61.03	<0.001 *
200–400 g/day	21.89	18.25	25.21	
>400 g/day	8.23	2.19	13.75	
Vegetable intake				
<200 g/day	13.28	25.22	2.36	<0.001 *
200–400 g/day	39.51	52.39	27.72	
>400 g/day	47.21	22.40	69.91	
**Biochemical index**				
HbAc1 (%)	9.95 (2.45)	9.90 (2.42)	9.99 (2.48)	0.375
UA (μmol/L)	325.55 (95.57)	330.42 (100.38)	321.13 (90.79)	0.013 *
ALT (U/L)	22 (15–35)	23 (15–38)	21 (15–33)	0.005 *
AST (U/L)	21 (17–28)	21 (17–29)	21 (17–27)	0.196
TGs (mmol/L)	1.52 (1.04–2.31)	1.55 (1.05–2.47)	1.51 (1.03–2.20)	0.003 *
TC (mmol/L)	4.94 (1.48)	4.98 (1.55)	4.91 (1.41)	0.198
HDL-c (mmol/L)	0.99 (0.85–1.16)	0.98 (0.83–1.14)	1.00 (0.86–1.18)	0.001 *
LDL-c (mmol/L)	2.71 (0.91)	2.72 (0.93)	2.70 (0.89)	0.488
HOMA-IR	2.35 (1.41–3.98)	2.46 (1.47–4.17)	2.24 (1.35–3.80)	0.008 *

Abbreviations: NAFLD: nonalcoholic fatty liver disease, DBP: diastolic blood pressure, SBP: systolic blood pressure, BMI: body mass index, WC: waist circumference, HC: hip circumference, HbAc1: glycosylated hemoglobin, UA: uric acid, ALT: alanine aminotransferase, AST: aspartate aminotransferase, TGs: triglycerides, TC: total cholesterol, HDL-c: high-density lipoprotein cholesterol, LDL-c: low-density lipoprotein cholesterol, HOMA-IR: homeostasis model assessment of insulin resistance. * The difference was significant between <500 g/day group and ≥500 g/day group (*p* < 0.05).

**Table 2 nutrients-14-04559-t002:** Odds ratios with 95% confidence intervals for nonalcoholic fatty liver disease according to daily fruit–vegetable intake.

	Cases/Participants (n/n, %)	Model 1		Model 2		Model 3		Model 4	
		OR (95% CI)	*p*	OR (95% CI)	*p*	OR (95% CI)	*p*	OR (95% CI)	*p*
**Fruit–vegetable intake**									
<500 g/day	714/1274 (56.04)	Ref		Ref		Ref		Ref	
≥500 g/day	731/1393 (52.47)	0.91 (0.75, 1.09)	0.302	0.90 (0.75, 1.09)	0.296	0.89 (0.73, 1.09)	0.257	0.89 (0.73, 1.09)	0.252
**Fruit intake**									
<200 g/day	976/1865 (52.33)	Ref		Ref		Ref		Ref	
200–400 g/day	338/583 (57.98)	1.23 (1.01, 1.49)	0.037 *	1.16 (0.96, 1.41)	0.130	1.10 (0.90, 1.36)	0.358	1.08 (0.88, 1.33)	0.462
>400 g/day	131/219 (59.82)	1.48 (1.10, 1.98)	0.009 *	1.39 (1.03, 1.86)	0.030 *	1.18 (0.86, 1.62)	0.316	1.15 (0.84, 1.59)	0.388
**Vegetable intake**									
<200 g/day	222/354 (62.71)	Ref		Ref		Ref		Ref	
200–400 g/day	580/1055 (54.98)	0.68 (0.53, 0.88)	0.004 *	0.72 (0.55, 0.93)	0.013 *	0.73 (0.56, 0.96)	0.027*	0.74 (0.56, 0.97)	0.032 *
>400 g/day	643/1258 (51.11)	0.59 (0.46, 0.75)	0.000 *	0.65 (0.50, 0.84)	0.001 *	0.66 (0.50, 0.86)	0.002*	0.67 (0.51, 0.88)	0.004 *

Model 1: Not adjusted; Model 2: Adjusted for demographic parameters (age, sex, educational attainment), anthropometric parameters (BMI, WC, HC, BP) and diabetes duration; Model 3: Adjusted for the variables in Model 2 plus family history, lifestyle factors (smoking, drinking, physical activity level) and food intake (the consumption of bean products, salt, fish and sugary beverages); Model 4: Adjusted for the variables in Model 3 plus biochemical index values (HbA1c, ALT, AST and serum lipid levels). * The difference was significant compared to reference groups (*p* < 0.05).

**Table 3 nutrients-14-04559-t003:** Subgroups analyses for the prevalence of nonalcoholic fatty liver disease among 2667 participants with the highest vs. lowest intake of vegetables.

	Cases/Participants (%)	OR (95% CI)	*p*	*p*-Interaction
**Sex**				0.755
Male	954/1692 (56.38)	0.69 (0.49, 0.96)	0.030	
Female	491/957 (50.36)	0.57 (0.34, 0.95)	0.029	
**Age**				0.640
<50 years	706/1181 (59.78)	0.74 (0.50, 1.10)	0.135	
≥50 years	739/1486 (49.73)	0.59 (0.40, 0.87)	0.008	
**Duration**				0.151
<67 months	766/1310 (58.47)	0.56 (0.38, 0.81)	0.002	
≥67 months	669/1341 (49.89)	0.83 (0.54, 1.25)	0.366	
**Overweight/Obesity**				0.598
No	485/1223 (39.66)	0.70 (0.47, 1.05)	0.088	
Yes	948/1423 (66.62)	0.64 (0.44, 0.94)	0.024	
**Central obesity**				0.432
No	552/1288 (42.86)	0.73 (0.48, 1.09)	0.125	
Yes	877/1350 (64.96)	0.63 (0.43, 0.92)	0.016	
**Hypertension**				0.201
No	556/1121 (49.6)	0.84 (0.55, 1.29)	0.433	
Yes	889/1546 (57.5)	0.56 (0.39, 0.81)	0.002	
**Hyperlipidemia**				0.168
No	472/1013 (46.59)	0.73 (0.45, 1.17)	0.191	
Yes	973/1654 (58.83)	0.62 (0.44, 0.87)	0.005	

All analyses were adjusted for demographic parameters, diabetes duration, anthropometric parameters, family history, lifestyle factors, food intake and biochemical indices. *p* values were calculated by logistic regression.

**Table 4 nutrients-14-04559-t004:** Standard β coefficient with 95% confidence intervals for the fatty liver index according to daily fruit and vegetable intake.

	Model 1		Model 2	
	β (95% CI)	*p*	β (95% CI)	*p*
Fruits	1.007 (−0.703, 2.717)	0.248	−0.255 (−1.137, 0.627)	0.571
Vegetables	−4.210 (−5.776, −2.644)	<0.001 *	−1.028 (−1.836, −0.219)	0.013 *

Model 1: Not adjusted; Model 2: Adjusted for demographic parameters, diabetes duration, anthropometric parameters, diabetes family history, lifestyle factors, food intake and biochemical index values (HbA1c, ALT, AST and serum lipid levels). * The β value and 95% CI was statistical significance (*p* < 0.05).

## Data Availability

The data of this study are available upon request to the corresponding author.

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
