# Peer review of "Inverse Association of Fruit and Vegetable Consumption with Nonalcoholic Fatty Liver Disease in Chinese Patients with Type 2 Diabetes Mellitus"

_nutrients, 2022, doi:10.3390/nu14214559_

Round 1

Reviewer 1 Report

Title: Inverse association of fruit and vegetable consumption with nonalcoholic fatty liver disease in Chinese patients with type 2 diabetes mellitus 

Although study looks interesting there are some issues with this manuscript.

1. English language throughout the manuscript needs to be improved.

2. Introduction:

-I recommend begin the introduction with a brief description of diabetes, linking NAFLD to obesity, insulin resistance, and type 2 diabetes. Answer the reader's question on the role of insulin in fat disposition and its correlation to NAFLD

- The authors should clarify the novelty of this article in the ‘Introduction’ and ‘Conclusion’ section.

3. Materials and Methods

-Authors need to include the manufacturer, supplier, country, and city for the reagents/instruments used for these studies.

4. Results:

-I would suggest to include the significant symbol (*) within the values.

5. Discussion:

-How is this article more informative than the previously published ones? Justify it.

- I would suggest to include the future prospective of these studies.

6.Conclusion:

-This section sounds good; however, I request to elaborate a bit more to make it easier to understand the reader.

Reviewer 2 Report

It would be useful if the Authors may supply (as supplementary material or even only for the Reviewer) the actual FFQ used. At present, it is unclear which data on whole diet were obtained, the Authors mention "fish and sugary beverage) in the Methods section, but never refer again to these two items in the Results or Discussion. They also state that in model 2 they adjusted the data for "food intake" but do not specificy if they mean total intake (calories?) or a specific dietetic pattern
